# Sniffer Dogs Diagnose Lung Cancer by Recognition of Exhaled Gases: Using Breathing Target Samples to Train Dogs Has a Higher Diagnostic Rate Than Using Lung Cancer Tissue Samples or Urine Samples

**DOI:** 10.3390/cancers15041234

**Published:** 2023-02-15

**Authors:** Shih-Feng Liu, Hung-I Lu, Wei-Lien Chi, Guan-Heng Liu, Ho-Chang Kuo

**Affiliations:** 1Division of Pulmonary and Critical Care Medicine, Department of Internal Medicine, Kaohsiung Chang Gung Memorial Hospital, Kaohsiung 833, Taiwan; 2Department of Respiratory Therapy, Kaohsiung Chang Gung Memorial Hospital, Kaohsiung 833, Taiwan; 3Medical Department, College of Medicine, Chang Gung University, Taoyuan 333, Taiwan; 4Department of Thoracic and Cardiovascular Surgery, Kaohsiung Chang Gung Memorial Hospital, Kaohsiung 833, Taiwan; 5Department of Post-Baccalaureate Veterinary Medicine, Asia University, Taichung 413, Taiwan; 6Department of Electronic Engineering, Chang Gung University, Taoyuan 333, Taiwan; 7Department of Pediatrics, Kaohsiung Chang Gung Memorial Hospital, Kaohsiung 833, Taiwan

**Keywords:** detection dog, lung cancer, breathing target samples

## Abstract

**Simple Summary:**

Sniffer dogs can diagnose lung cancer. However, the diagnostic yields of different samples and training methods for lung cancer remain undetermined. Six dogs were trained in three stages with the aim of improving the diagnostic yield. The findings showed the dogs had a very low response rate to urine target samples in the first and second stages training. Using exhaled air samples for training them to recognize exhaled samples, the diagnosis rate in lung cancer patients was 71.3% to 97.6% (mean 83.9%), whereas the false positive rate of lung cancer in the healthy group was 0.5% to 27.6% (mean 7.6%). The sensitivity and specificity of using exhaled breath target training (91.7% and 85.1%) were higher than using lung cancer tissue training (50.4% and 50.1%). Additionally, sniffer dogs diagnose lung cancer, independent of staging, pathologic type, and tumor location. Using sniffer dogs to screen human lung cancer may have clinical potential.

**Abstract:**

Introduction: Sniffer dogs can diagnose lung cancer. However, the diagnostic yields of different samples and training methods for lung cancer remain undetermined. Objective: Six dogs were trained in three stages with the aim of improving the diagnostic yield of lung cancer by comparing training methods and specimens. Methods: The pathological tissues of 53 lung cancer patients and 6 non-lung cancer patients in the Department of Thoracic Surgery of Kaohsiung Chang Gung Hospital were collected, and the exhaled breath samples and urine samples were collected. Urine and exhaled breath samples were also collected from 20 healthy individuals. The specimens were sent to the Veterinary Department of Pingtung University of Science and Technology. Results: The dogs had a very low response rate to urine target samples in the first and second stages of training. The experimental results at the second stage of training found that after lung cancer tissue training, dogs were less likely to recognize lung cancer and healthy controls than through breath target training: the response rate to exhaled breathing target samples was about 8–55%; for urine target samples, it was only about 5–30%. When using exhaled air samples for training, the diagnosis rate of these dogs in lung cancer patients was 71.3% to 97.6% (mean 83.9%), while the false positive rate of lung cancer in the healthy group was 0.5% to 27.6% (mean 7.6%). Compared with using breathing target samples for training, the diagnosis rate of dogs trained with lung cancer tissue lung cancer was significantly lower (*p* < 0.05). The sensitivity and specificity of lung cancer tissue training (50.4% and 50.1%) were lower than the exhaled breath target training (91.7% and 85.1%). There is no difference in lung cancer diagnostic rate by sniff dogs among lung cancer histological types, location, and staging. Conclusion: Training dogs using breathing target samples to train dogs then to recognize exhaled samples had a higher diagnostic rate than training using lung cancer tissue samples or urine samples. Dogs had a very low response rate to urine samples in our study. Six canines were trained on lung cancer tissues and breathing target samples of lung cancer patients, then the diagnostic rate of the recognition of exhaled breath of lung cancer and non-lung cancer patients were compared. When using exhaled air samples for training, the diagnosis rate of these dogs in lung cancer patients was 71.3% to 97.6% (mean 83.9%), while the false positive rate of lung cancer in the healthy group was 0.5% to 27.6% (mean 7.6%). There was a significant difference in the average diagnosis rate of individual dog and overall dogs between the lung cancer group and the healthy group (*p* < 0.05). When using lung cancer tissue samples for training, lung cancer diagnosis rate of these dogs among lung cancer patients was only 15.5% to 40.9% (mean 27.7%). Compared with using breathing target samples for training, the diagnosis rate of dogs trained with lung cancer tissue lung cancer was significantly lower (*p* < 0.05). The sensitivity and specificity of lung cancer tissue training (50.4% and 50.1%) were lower than the exhaled breath target training (91.7% and 85.1%). The diagnostic rate of lung cancer by sniffer dogs has nothing to do with the current stage of lung cancer, pathologic type, and the location of tumor mass. Even in stage IA lung cancer, well-trained dogs can have a diagnostic rate of 100%. Using sniffer dogs to screen early lung cancer may have good clinical and economic benefits.

## 1. Introduction

Lung cancer is the leading cause of cancer mortality. Early detection and surgical removal of lung tumors is the best action for long-term survival or cure, but it is only suitable for 15% to 20% of lung cancer populations [1]. When the disease is discovered at an advanced stage, treatment results are usually unsatisfactory breathing target samples [2]; often, lung cancer develops asymptomatically and makes early diagnosis difficult. Furthermore, existing screening methods cannot effectively distinguish between individuals who have lung cancer and those who do not [3]. Breath sampling may provide a potentially useful method because evidence demonstrates that unique chemical characteristics can be detected in the lungs of patients with lung cancer and that exhaled respiratory biomarkers can help with clinical decision making [4,5].

Dogs have an extremely sensitive sense of smell and can distinguish an array of odors. In fact, the resolution capacity of dogs’ olfaction is higher than the best current technologies [6]. There are already many applications of dog detection [7], including the uncovering of abiotic substances for narcotic blasting explosives, land mines, arson materials, pirated discs, DVDs, and illegal currency, among others. Biological uses include many ecological conservation projects to assist wildlife research or pest control, such as the search of wild animal waste and the search for animal bodies, individuals, alien species, and pests. Search-and-rescue and cadaver-detection dogs are also widely used in human-related detection. For human diseases, dogs are trained to detect signs of epileptic seizures and cancer [8,9,10,11,12,13,14,15,16,17,18,19,20]. Cancer detection by a dog was first proposed by Williams in 1989, and it is believed that dogs can detect the odor emitted by malignant tumors [11]. An early study published in 2006 on the detection of lung and breast cancer investigated this phenomenon [14]. Samples were taken from 55 lung cancer patients, 31 breast cancer patients, and 83 healthy patients. The gas exhaled by these patients contained a substance that was not detected by the instrument used. Canine detection was subsequently performed using a double-blind test. The sensitivity and specificity of the lung cancer group reached 0.99, while those of the cancer group were 0.88 and 0.98, respectively. The sensitivity and specificity of the two cancers in each cancer stage were similar; hence, the detection efficiency of the dogs was very high [14]. In another study from 2011, two German wolves, one Australian Shepherd, and one Labrador were used. Two hundred and twenty volunteers provided breath samples for testing, and the dogs were trained to sniff test tubes in a positive and encouraging manner. The test tube was touched to the snout, and each sample was used only once. The detected diseases were lung cancer and chronic obstructive pulmonary disease, with a sensitivity of 71% and a specificity of 93% [17].

Following these studies, researchers have expanded on canine detection of suspected lung cancer in non-selective patients to include their urine and exhaled gases [18]. The use of canine detection for lung cancer screening has shown significant potential and has been the subject of clinical studies in the recent years [20,21]. A multidisciplinary approach, involving surgeons, radiation oncologists, pulmonologists, and oncologists, is required to optimize the survival and quality of life of patients with lung cancer [22]. At present, the samples provided for dog olfactory training are mainly lung cancer tissue and exhaled gas samples. However, the diagnostic yields of different samples and training methods for lung cancer remain undetermined. This study aims to find better training methods and samples to improve the diagnosis rate of lung cancer.

## 2. Material and Methods

### 2.1. Study Design

We collected specimens from study participants, including lung cancer patients, non-lung cancer patients, and healthy volunteers, from the Department of Thoracic Surgery, Chang Gung Hospital, Kaohsiung, Taiwan. The samples included pathological tissue, exhaled breath samples, and urine samples, and were frozenly transported to the Department of Veterinary Medicine, Pingtung University of Science and Technology, Taiwan. Healthy controls only sent urine and exhaled breath samples to the study center. Six dogs were trained in three stages with the aim of improving the diagnostic yield of lung cancer by comparing training methods and specimens. The study was conducted from 3 May 2016 to 31 July 2017.

### 2.2. Study Participants

The study enrolled a total of 79 subjects, including 53 patients with lung cancer, 6 patients with benign lung diseases, and 20 healthy controls. The diagnosis of lung cancer patients was confirmed by preoperative clinician judgment, imaging studies, bronchoscopic biopsy pathology report, and postoperative pathology report. Non-lung cancer benign patients are clinically unconfirmed before operation but confirmed by pathology report after operation. The healthy controls were volunteers aged 18 to 35 years old, who had normal chest radiography, no history of smoking, and no clinical history of malignancy.

### 2.3. Target and Non-Target Samples

Target samples included lung cancer tumor tissues, breathing target (BT), and urine target (UT) samples. Non-target samples included breathing non-target and urine non-target samples. Breathing non-target samples were collected from either healthy controls (BNT) or patients with a confirmed diagnosis of non-lung cancer (BTN). Urine non-target samples were also collected from healthy controls (UNT) or patients with a confirmed diagnosis of non-lung cancer (UTN). Five exhaled breath samples and five urine samples were collected from each enrolled subject, respectively.

### 2.4. Preparation of Lung Cancer and Non-Lung Cancer Tissue Samples

In this study, we used the freeze-drying method on the lung cancer and non-lung cancer tissue samples to prevent deterioration from high temperature exposure. The raw tissues were directly freeze-dried to eliminate the moisture of the lung samples. Alcohol- or formalin-fixed specimens were not used to avoid affecting the smell of the detection dog.

### 2.5. Preparation of Exhaled Breath and Urine Samples

Both exhaled breath and urine used for the olfactory test were collected from participants at least eight hours after oral intake or tobacco smoking. A filter paper roll was placed in an air collection tube, of diameter 12 mm and length 7 cm, to collect the exhaled gas of the patient. The lid was screwed tightly, stored in a refrigerated environment at 4 °C, and transported to the dog detection center. These filters were placed into the tubes at room temperature one hour before the olfactory test. Additionally, 10 cc of urine was collected, placed in a freezing tube, and stored at −20 °C for refrigerated transportation to the detection dog center. Urine was thawed one hour before the olfactory test and immediately placed in a separate, specially sealed glass container to prevent different odorous compounds from mixing.

### 2.6. Dog Training

Dog training was conducted at the Department of Veterinary Medicine, Pingtung University of Science and Technology, Taiwan. An established procedure was used to train the dogs; there were no training differences between them. Dogs were trained by a team including a veterinarian and 3 helpers. The selected six dogs are all well-trained, and they have all been involved in the detection of brown root disease of road trees or anti-drug work. The first stage of training for lung cancer identification took about 4 to 6 months. Lung cancer detection in six dogs begins after about four months of training. Considering the quality of specimen storage, our specimens are sent to Ping University of Science and Technology one after another with the collection of cases, not at one time. Most of the specimens delivered earlier are used to train dogs, and the specimens that arrive later are used in the detection of lung cancer.

### 2.7. Training Method

Unmarked target (lung tumor tissue) and non-target (control) samples were randomly placed in unlabeled jars. The test was video recorded, and a technical check to ensure accurate recording was first carried out. After the target was in position, the dog leader brought out the dog for detection. If there was a reaction to the target, a positive response was recorded; otherwise, a negative response was recorded. Response to non-targets was recorded as a false positive. Dogs were trained to sit down as a reaction and were given a food reward if the reaction was positive (Figure 1). To maintain the high sensitivity of the olfactory test, the dog was trained at least twice a week for the duration of the study.

### 2.8. First Stage of Training

At this stage, all six dogs learned the smell of lung cancer. Training using three different target samples (tumor tissue, exhalation breath, and urine) was completed for the cognitive association of lung cancer odor and was ready for follow-up tests.

### 2.9. Second Stage of Training

The purpose of the experiment at this stage was to observe the canine response rate to the target and non-target breath/urine samples after receiving detection training using lung cancer tissue. We led the dogs to identify the BT and UT, and later, to recognize TNT, BNT, and UNT samples; we recorded these results accordingly.

### 2.10. Third Stage of Training

The purpose of the experiment at this stage was to observe the canine response rate to the target and non-target breath/urine samples after receiving training using breathing target samples.

### 2.11. Ethical Issue

The study was approved by the Chang-Gung Memorial Hospital review board (IRB # 105-1513C) on 3 May 2016. Written informed consent was obtained from each patient. The 6 dogs used were of the Miglu breed. The dogs were service dogs in-training. They were all trained detection dogs that were already performing tasks. They were originally used for insect detection. We obtained verbal consent from the trainer team prior to this study. During this experiment period, only the target items were added. There is no drug administration or invasive harm to the body in this study. These dogs are cared for by the Department of Veterinary Medicine of Pingtung Institute of Technology. They are cared for by the dedicated persons and provided with excellent housing and eating conditions. General medical care is taken care of by veterinarians. In addition, for this research, we have provided some funds to the trainer team for dog housing conditions, enrichment activities, and general medical care. The patients’ samples were sent from the hospital to the school, and the school completed the study. The dogs remained in school to perform their original task after this study.

### 2.12. Statistical Analysis

We use Statistical software SPSS WP 17 (SPSS Inc. (Chicago, IL, USA), 2008) for descriptive statistics and frequency calculations. Sensitivity and specificity were used to describe the accuracy of the tests. The sensitivity (true positive rate) was calculated as the proportion of cancer samples correctly recognized by the dog, and the specificity (true negative rate) was calculated as the ratio of the number of cancer samples detected to the total number of individuals in the cancer negative control group. Using the binomial probability distribution and a conventional two-by-two (2 × 2) table, sensitivity was generally defined by the equation a/(a + c) and specificity was defined by d/(b + d). The individual and total average lung cancer diagnostic rate of 6 dogs trained using exhaled breathing target samples between lung cancer group and healthy subjects group and comparison of the diagnostic rate between training dogs using breathing target samples then the recognition of exhaled gases samples and training dogs using lung cancer samples then the recognition of exhaled gases samples in lung cancer group use t-test. A *p* value < 0.05 was considered significant.

## 3. Results

### 3.1. The Characteristics of Participants

Table 1 shows the characteristics of participants enrolled in the study. The 79 total subjects included 53 lung cancer patients (67.1%), 6 non-lung cancer patients (7.6%), and 20 healthy controls (25.3%). Of the lung cancer patients, 46 had adenocarcinoma. Pathology of non-lung cancer subjects (N = 6) included anthracosis (N = 1), fibrosis (N = 1), necrotizing granulomatous inflammation (N = 2), pulmonary chondroid hamartoma (N = 1), and chronic inflammation (N = 1), which also ruled out malignancy clinically.

### 3.2. Results of the Second Stage Training

Table 2 shows the individual and total average lung cancer diagnostic rates of the dogs trained using lung cancer tissue samples; the rates of discriminating lung cancer and non-lung cancer using exhaled gas and urine samples were also reported. The rate of lung cancer diagnosis through the recognition of exhaled gases ranged from 15.5% to 40.9% (average range), with an average of 27.7% in lung cancer patients. Using exhaled gases, lung cancer recognition after tissue training had a sensitivity of 50.4% and a specificity of 50.1%. Figure 2 shows individual canine response rates to exhaled breath (Figure 2A) and urine samples (Figure 2B) collected from patients with a confirmed diagnosis of lung cancer, from patients with a confirmed diagnosis of non-lung cancer, and from healthy controls after using lung cancer tissue training. The experimental results at the second stage of training found that after lung cancer tissue training, dogs recognize lung cancer and healthy controls more than through breath training: the response rate to exhaled breathing target samples was about 8–55%; for urine target samples, it was only about 5–30%.

### 3.3. Results of the Third Stage of Training

Table 3 shows that the individual and total average lung cancer canine diagnostic rates were significantly different between the lung cancer patients and the healthy subjects when training dogs with exhaled breathing target samples. The average positive rate of detecting lung cancer was between 71.3% and 97.6% when using samples from lung cancer patients, a range much higher than the average false-positive rate of 0.5% to 27.1% when using samples from healthy controls. The sensitivity and specificity using exhaled breathing target samples was 91.7% and 85.1%, respectively. Figure 3 depicts the individual canine response rate (N = 6) to exhaled breathing samples from cancer patients and healthy controls after training using this sample method. Although there were mild differences between individuals, the detection rate was not related to tumor location, lung cancer staging, or pathology type. Table 4 shows that training dogs using exhaled breathing target samples had a higher diagnostic rate for lung cancer than training dogs using lung cancer samples in diagnosed patients. The sensitivity and specificity in dogs trained using exhaled breathing target samples (91.7% and 85.1%, respectively) were higher than those in training dogs using lung cancer samples (50.4% and 50.1%, respectively). The diagnostic rate of lung cancer by sniffer dogs has nothing to do with the current stage of lung cancer, pathologic type, and the location of tumor mass. The average diagnosis rate of each sniffer dog for 47 BT samples (stage IA) is lower than those of other patients, 13.3% in dog E, 40% in dog D, and 46.7% in dog F, but it can still be achieved in dog A and B at 86.7%. In the BT samples of other patients, dog A has a lung cancer diagnosis rate of 100% regardless of stage IA, IIA, IIIB, or different locations (Table 5).

## 4. Discussion

The experimental results at the second stage of training found that after lung cancer tissue training, dogs were less likely to recognize lung cancer and healthy controls than through breath training: the response rate to exhaled breathing target samples was about 8–55%; for urine target samples, it was only about 5–30%. Furthermore, the dogs had a very low response rate to urine target samples in the first and second stages training. Therefore, our study demonstrated training dogs using exhaled breathing target samples had a higher diagnostic success rate for lung cancer than training dogs using lung cancer samples. The sensitivity and specificity in dogs trained using exhaled breathing target samples (91.7% and 85.1% respectively) were higher than those in training dogs using lung cancer samples (50.4% and 50.1%, respectively). Using this method, there is no difference of lung cancer diagnostic rate by sniff dogs among lung cancer histological types, locations, and cancer stages. It is interesting that some smart well-trained dogs can diagnose lung cancer at different stages and even stage IA, even reaching 100%.

Training dogs using exhaled breathing target samples had a higher diagnostic success rate for lung cancer than training dogs using lung cancer samples. There are some reasons for this difference as follows: we used a drying method, and not chemical treatment, to preserve lung tissue samples in this study, as chemicals may interfere with canine odor detection. Although lung cancer tissue samples contain dense cancer cells, storing the samples dry may prevent the necessary evaporation of important volatile components, eliminating the ability of the dog to detect the odor. In contrast, frozen filter strips contain the components of the gas that are blown out of the patient’s underlying disease and can release the volatile component into the container when it melts at room temperature. Volatile organic compounds (VOCs) are very important to the differentiation of underlying disease by detecting dogs. Horváth et al. have demonstrated that exhaled biomarkers contain both volatile and nonvolatile molecules. The profile of VOCs in patients with lung cancer was different from those in control subjects in their study [5]. Chen et al. found that 11 characteristic VOCs were present in higher amounts in lung cancer patients than in control subjects, which included chronic bronchitis patients as well as healthy subjects [21]. They also found that tumor cells and cancer-type VOC profiles that appeared macroscopically normal in lung tissue samples were confirmed after 10 days post-cell cultivation. They considered that profound changes occur during the initial stages of carcinogenesis by altering VOC production/elimination in the tumor microenvironment; therefore, VOC analysis may detect lung cancer at the earliest stages.

Using exhaled breathing target samples to train dogs has some advantages when compared with using lung cancer tissue or urine samples. Exhaled breath samples are easy to obtain, while lung cancer or other body tissue samples are more difficult to acquire and so may have limited sample sizes. The current method of preserving exhaled air is by using filter paper, but we look forward to a better storage method in the future. Lung cancer tissue samples were stored using a dry method in this study. If stored with chemicals, the odor of chemicals might affect the dogs’ olfactory training. In addition, exhaled breath and gas sampling is completely noninvasive, hence it is more in line with the possibility of large-scale clinical screening for lung cancer [5].

Furthermore, the use of urine samples for training dogs to detect lung cancer has previously been reported [18,23]. Exhaled gas from cancerous lungs contains specific substances, which are theoretically feasible for dogs to recognize through exhaled breath samples. However, whether lung disease or different stages of lung cancer cause specific substances in the urine needs to be further explored. In our study, dogs had a lower response rate to urine target samples when compared with exhaled breathing samples in the second stage of training, so we did not use urine for subsequent training stages. In addition, the urine composition may be affected by food, drugs, or underlying diseases, as well as the external environment and time duration. For large-scale lung cancer screening and clinical practice, urine samples are less convenient and effective than exhaled gas samples.

There is a lot of room for future development in the diagnosis of lung cancer via canine detection. Using trained dogs to diagnose lung cancer as a clinical screening tool is cost-effective and allows for the diagnosis and treatment of lung cancer patients at an early stage [18,19,20]. However, while canine detection is used in a large number of lung cancer screening tools, further confirmation is required through imaging studies, pathological examination, or cytological study. A study by Hackner et al. revealed final positive and negative predictive values of 30.9% and 84.0%, respectively, which were lower than those reported in other studies. These low results were explained by the lack of a positive response from the dog and operator, which was thought to cause higher levels of canine stress [24]. Nevertheless, many scholars recognize that the diagnosis of lung cancer by dogs has a high accuracy rate. Until a sensitive and effective electronic nose is invented, future canine-related research will continue [25,26]. In addition, we look forward to utilizing further scientific research methods to identify unique substances exhaled by lung cancer patients and to invent clinically applicable instruments able to detect them. In one study, gas chromatography–mass spectrometry (GC-MS) was used to analyze and compare exhaled gas with the result from canine detection [27]. In their study, the authors found that there was a positive correlation between dog indications and the ethyl acetate and 2-pentanone content of breath (r = 0.85 and r = 0.97, respectively). Rudnicka et al. also describe the names of particular volatile organic compounds that potentially serve as biomarkers of lung cancer and of course proves that dogs can be used to detect such disease with overall sensitivity of canine scent detection about 86%, and specificity about 72% [28]. Additionally, canine detection may be applied before and after lung cancer surgery. In this case, dogs can recognize the disappearance of tumors after surgery and may be utilized for complete assessment of surgery or follow-up.

There are several strengths to using dogs to diagnose lung cancer. First, large-scale community screening is available for early diagnosis and early treatment of lung cancer. Second, the procedure is simple and convenient. Third, this method may reduce medical expenses. However, there are also some disadvantages or limitations. First, although this study attempted to maintain the integrity of the original taste of the samples, during the implementation process, the canine diagnosis may be affected by the pollution of the surrounding environment. Second, the public lacks confidence in using dogs to diagnose lung cancer, and more medical consensus needs to be established [29,30]. Third, different dogs have different diagnostic sensitivities for lung cancer. Dogs need to be certified. Forth, dogs may have emotions, and the consistency of daily on-duty diagnosis needs to be considered [29]. Therefore, after the dog is diagnosed, it is still necessary to follow the current medical process to diagnose lung cancer.

## 5. Conclusions

Training dogs using breathing target samples to train dogs then to recognize exhaled samples had a higher diagnostic rate than training using lung cancer tissue samples or urine samples. Using this method, sniffer dogs diagnose lung cancer, independent of lung cancer stage, pathologic type, and tumor location. Dogs had a very low response rate to urine samples in our study.

## Figures and Tables

**Figure 1 cancers-15-01234-f001:**
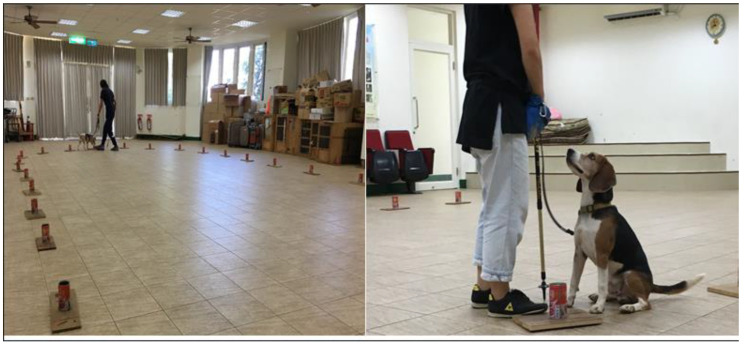
Dogs training. The target was randomly placed in a jar, and the dog leader brought the dog forward for detection. If there was a reaction to the target, a positive response was recorded; otherwise, a negative response was recorded. Response to non-targets was recorded as a false positive. Dogs were trained to sit down as a reaction and given a food reward if the reaction was positive.

**Figure 2 cancers-15-01234-f002:**
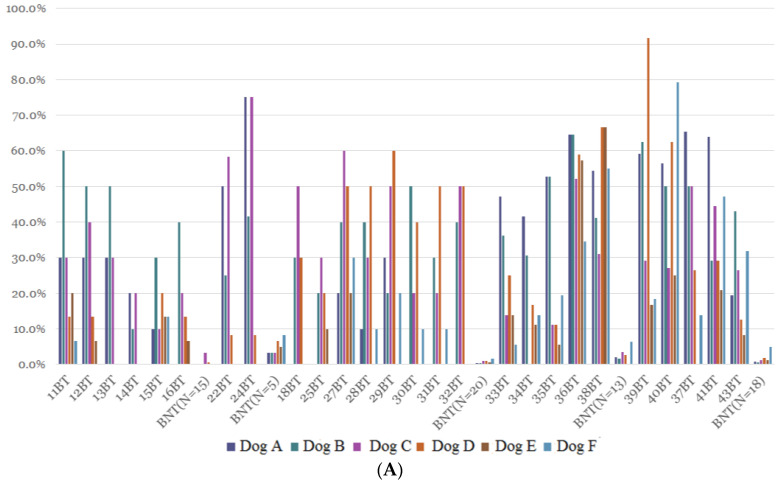
(**A**) Individual canine response rates after using lung cancer tissue samples training. Individual canine response rates to exhaled breath collected from patients with a confirmed diagnosis of lung cancer, from patients with confirmed diagnosis of non-lung cancer, and from healthy controls after using lung cancer tissue samples training. (**B**) Individual canine response rates after using lung cancer tissue samples training. Individual canine response rates to urine samples collected from patients with a confirmed diagnosis of lung cancer, from patients with confirmed diagnosis of non-lung cancer, and from healthy controls after using lung cancer tissue samples training.

**Figure 3 cancers-15-01234-f003:**
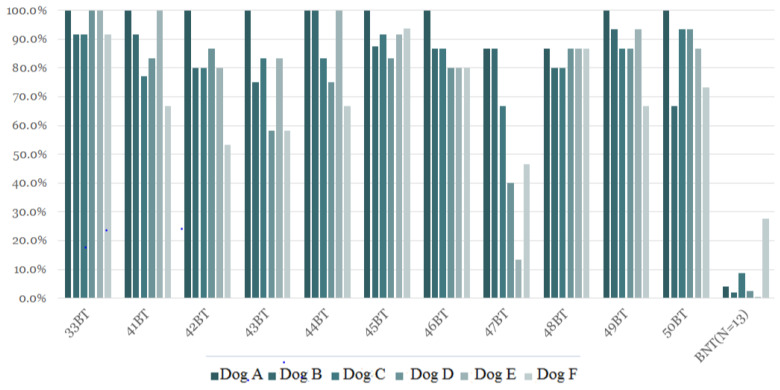
Individual canine response rates after using breathing target training. Individual canine response rates to exhaled breathing samples collected from patients with a confirmed diagnosis of lung cancer and from healthy controls after using breathing target training.

**Table 1 cancers-15-01234-t001:** Characteristics of enrolled participated subjects in the study.

	Lung Cancer N (53)	Non-Lung Cancer N (6)	Healthy Controls N (20)
Total subjects (N = 79)	53 (67.1%)	6 (7.6%)	20 (25.3%)
Age (years ± SD)	63.3 ± 9.5	56.6 ± 6.3	29.2 ± 2.5
Gender			
Male	31 (55.4%)	1 (16.7%)	8 (40%)
Female	24 (44.6%)	5 (83.3%)	12 (60%)
Lung cancer (N = 53)			
Adenocarcinoma	46 (58.2%)		
Squamous cell carcinoma	5 (6.3%)		
Small cell lung cancer	1 (1.3%)		
Spindle cell carcinoma	1 (1.3%)		
Staging of lung cancer			
Stage I	28 (35.4%)		
Stage II	10 (12.7%)		
Stage III	10 (12.7%)		
Stage IV	5 (6.3%)		
Location of tumor			
RUL	16 (30.2%)	2 (33.2%)	
RML	1 (1.9%)	1 (16.7%)	
RLL	8 (15.1%)	1 (16.7%)	
LUL	21 (39.6%)	1 (16.7%)	
LLL	7 (13.2%)	1 (16.7%)	

Pathology of non-lung cancer subjects (N = 6) including anthracosis (N = 1), fibrosis (N = 1) necrotizing granulomatous inflammation (N = 2), pulmonary chondroid hamartoma (N = 1), and chronic inflammation (N = 1), which also ruled out malignancy clinically. RUL: right upper lobe; RML: right middle lobe; RLL: right lower lobe, LUL: left upper lobe; LLL: left lower lobe.

**Table 2 cancers-15-01234-t002:** The individual and total average lung cancer diagnostic rate of six dogs using lung cancer samples training and then the recognition of exhaled gases and urine samples in lung cancer and non-lung cancer patients groups was compared in the second stage training.

	Dog A	Dog B	Dog C	Dog D	Dog E	Dog F	Average
BT (N = 14) (%)	28.9	40.9	32.0	33.4	15.9	15.5	27.7
BTN (N = 12) (%)	35.4	38.6	35.9	29.9	6.6	16.8	27.2
BNT (N = 71) (%)	0.9	0.8	2.1	1.9	0.9	3.5	1.7
UT (N = 13) (%)	15.9	23.7	30.9	17.8	11.7	17.4	19.5
UTN (N = 9) (%)	7.8	8.2	14.3	16.3	2.0	16.1	10.8
UNT (N = 68) (%)	1	1.2	2.16	4.06	1.06	4.6	2.3

Breathing Target (BT): exhaled breathing samples collected from the patients with confirmed diagnosis of lung cancer. BTN: exhaled breathing samples collected from the patients with confirmed diagnosis of non-lung cancer. Breathing Non-target (BNT): exhaled breathing samples collected from healthy controls. Urine Target (UT): urine samples collected from the patients with confirmed diagnosis of lung cancer. UTN: urine samples collected from the patients with confirmed diagnosis of non-lung cancer. Urine Non-target (UNT): urine samples collected from healthy controls.

**Table 3 cancers-15-01234-t003:** The individual and total average lung cancer diagnostic rate of six dogs trained using exhaled breathing target samples are significantly different between the lung cancer group and healthy subjects group.

	Dog A	Dog B	Dog C	Dog D	Dog E	Dog F	Average
BT (N = 11) (%)	97.6 *	85.4 *	86.7 *	79.4 *	83.2 *	71.3 *	83.9 *
BNT (N = 13) (%)	4.1	2.1	8.7	2.6	0.5	27.7	7.6

* BT vs. BNT, *p* < 0.05. Breathing Target (BT): exhaled breathing samples collected from the patients with confirmed diagnosis of lung cancer. Breathing Non-target (BNT): exhaled breathing samples collected from healthy controls.

**Table 4 cancers-15-01234-t004:** Training dogs using exhaled breathing target samples has a higher diagnostic rate for lung cancer than training dogs using lung cancer samples in lung cancer group.

	A	B	C	D	E	F	Average
BT ^1^ (N = 11) (%)	97.6 *	85.4 *	86.7 *	79.4 *	83.2 *	71.3 *	83.9 *
BT ^2^ (N = 14) (%)	28.9	40.9	32.0	33.4	15.9	15.5	27.7

* BT ^1^ vs. BT ^2^, *p* < 0.05. Breathing Target (BT): exhaled breathing samples collected from the patients with confirmed diagnosis of lung cancer. BT ^1^: Training dogs using breathing target samples then the recognition of exhaled gases samples in lung cancer group. BT ^2^: Training dogs using lung cancer samples then the recognition of exhaled gases samples in lung cancer group.

**Table 5 cancers-15-01234-t005:** The relationship between the diagnostic rates of lung cancer by sniffer dog using BT and lung cancer stage or tumor mass.

Sample	Lung Cancer Stage	Location	Dog A	Dog B	Dog C	Dog D	Dog E	Dog F
33 BT	Adenocarcinoma (IIIB)	RUL	100%	91.7%	91.7%	100%	100%	91.7%
41 BT	Adenocarcinoma (IA)	LUL	100%	91.7%	77.1%	83.3%	100%	66.7%
42 BT	Squamous cell carcinoma (IA)	LUL	100%	80.0%	80.0%	86.7%	80.0%	53.3%
43 BT	Adenocarcinoma (IIIA)	RUL	100%	75.0%	83.3%	58.3%	83.3	58.3%
44 BT	Adenocarcinoma (IA)	LUL	100%	100%	83.3%	75.0%	100%	66.7%
45 BT	Adenocarcinoma (IA)	LLL	100%	87.5%	91.7%	83.3%	92.7%	93.8%
46 BT	Adenocarcinoma (IIA)	LUL	100%	86.7%	86.7%	80.0%	80.0%	80.0%
47 BT	Adenocarcinoma (IA)	RLL	86.7%	86.7%	66.7%	40.0%	13.3%	46.7%
48 BT	Adenocarcinoma (IA)	LLL	86.7%	80%	80.0%	86.7%	86.7%	86.7%
49 BT	Adenocarcinoma (IA)	RUL	100%	93.3%	86.7%	86.7%	93.3%	66.7%
50 BT	Adenocarcinoma (IIIB)	LUL	100%	66.7%	93.3%	93.3%	86.7%	73.3%

Breathing Target (BT): exhaled breathing samples collected from the patients with confirmed diagnosis of lung cancer. RUL: right upper lobe; RML: right middle lobe; RLL: right lower lobe, LUL: left upper lobe; LLL: left lower lobe.

## Data Availability

Data supporting reported results can be obtained on request.

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
