# Peer review of "Sniffer Dogs Diagnose Lung Cancer by Recognition of Exhaled Gases: Using Breathing Target Samples to Train Dogs Has a Higher Diagnostic Rate Than Using Lung Cancer Tissue Samples or Urine Samples"

_cancers, 2023, doi:10.3390/cancers15041234_

Round 1

Reviewer 1 Report

It is an interesting study

it is not clear how trained dogs cah help to detect early stage lung cancer 

in patients who are already diagnosed for lung cancer.

I have some doubts about the significance of the study

I believe that dogs , having a developed sense of smell, can

detect the compounds in patients' exhaled breath, particularly of smokers compared to urine and tissue samples.

I suggest to include a reference about the interdisciplinariety for lung cancer diagnosis and management

-Oncol Lett. 2017 Sep;14(3):3035-3038.

Reviewer 2 Report

Dear Authors,

The article makes an interesting contribution to the field on use of sniffer dogs for the diagnosis of lung cancer in humans. However, the article requires significant changes in the structure of the text. It needs to be put in order, and requires some additions. I have comments before the manuscript could be considered for further proceeding and publication:

Abstract

1.      The abstract is not concise and requires correction in line with the adopted structure for the abstract of the scientific article, for instance: (1) Introduction, (2) Aim of the study, (3) Materials and Methods, (4) Results, and (5) Conclusions.

Introduction

2.      Introduction should be revised and corrected so the reader would be clearly informed about the rationale of the study: What was the background and motivation to conduct the study? What has not yet been explored in previous works on this topic?  

Aim of the study

3. The aim of the study should be identical in the abstract and in the main text.

Materials and Methods

4.      The Materials and Methods need to be organized better, and should be prepared in more understandable way. Please put in order, revise and re-edit information in such paragraphs: Study design, Study participants, Dog training, and describe them more precisely.

5.      On the beginning of the section a paragraph “2.1. Study design” should include information: on the type of the study that was carried out, where the study was conducted. Please, indicate the period and the dates, and general information about the study.

In lines 81-84 appears “urine samples”, what should be corrected.

6.      Then in2.2. Study participants” please, describe better, how the study participants and the control group were recruited, and also what where the inclusion to the study criteria. In addition, what diagnostic tests were carried out to confirm lung cancer in patients ?

7.      Paragraph „ 2.6. Dog training” requires more detailed explanation about how long the dogs were trained and how many people trained them.

8.      In paragraph 2.11 need to add the date of decision made by Chang-Gung Memorial Hospital review board.

9.      The paper does not describe how many breath and urine samples were collected during the total study period, i.e. at the stage of dog training and then at the next stage of the study. Please expand them.

10.  In statistical analysis, please add a level of p-value.

Results

11.  The results (in 3.1.) need to be given in a more detailed format - about patients and control group, especially about their age and sex.

12.Please explain abbreviations and meaning of the acronyms in the Tables 1 and 5 (RUL, RML, RLL, LUL, LLL).

Discussion

13.  The first paragraph in the Discussion should contain the more concluding sentences describing the „main findings”.

14.  The sentence in lines 271-272: “It has the potential to be clinically and economically beneficial in screening the population for lung cancerrequires a broader explanation based on the literature.

15.  Please add additional separated paragraph on the strengths and limitations of the study. Were there factors related to environmental conditions and its smells that could interfere with the test results at the stage of sample preparation and collection?

Conclusions

16. The Conclusions of the work should be more informative, with response to the purpose of the study (in the places regarding urine samples). Please expand that.

Tittle

17.  According to the reviewer, the title of the article is inadequate to the results, because there was no analysis on clinical and economic benefits. Please, correct that.

Please, highlight the changes to the revised version using a different colour.

Reviewer 3 Report

The authors investigate and found sniffer dogs to screen for early lung cancer .

The study is so interesting, however, I have some concerns to discuss.

-Specifically, please illustrate how a dog detects lung cancer by its sense of smell.

-Is there a difference between histological types in lung cancer?

-How do you use if patients dislike dogs?

Author Response

pleas see the attachment

Round 2

Reviewer 2 Report

Thank you for responding to my previous comments. I believe that the manuscript has improved considerably now. However, in my opinion, there are several issues that need to be corrected:

1.    The aim of the study does not fully refer to the topic of your research – please add information about dogs

2.    in 206 line is "Statistics", but should be "Statistical analysis"

3.    The Table 1 requires several additions:

·     lack units for variables, e.g. Age - this should be changed to "Age (years ± SD)"

·     information about women is missing – please add this line

· for RLL, LUL, LLL please complete the value of the percentage in parentheses

Reviewer 3 Report

The authors replied well, so the manuscript is suitable for publication.
